# Are Cities Saving Energy by Getting Smarter? Evidence from Smart City Pilots in China

Fei Xue [1],*, Minliang Zhou [2] and Jiaqi Liu [3]

1 Faculty of Applied Economics, University of Chinese Academy of Social Sciences, Beijing 102488, China
2 Institute of Industrial Economics, Chinese Academy of Social Sciences, Beijing 100006, China
3 School of Management, Xi'an University of Science and Technology, Xi'an 710699, China
* Correspondence: xuefei1@ucass.edu.cn

**Abstract:** Taking smart city pilots (SCP) in China as a quasi-experiment, this paper uses the staggered difference-in-differences (staggered DID) to examine the impact of the SCP policy on energy consumption by using panel data of 224 prefecture-level cities from 2006 to 2019. The results showed that the SCP policy reduces energy consumption and energy intensity by 3.3% and 5.3%, respectively. Heterogeneity analysis found that the energy-saving effect of the SCP policy is stronger in western cities, resource-based cities, and in cities that were the pioneering pilots. Mechanism analysis showed that smart industry transformation is the main transmission mechanism. Our findings have important practical implications for reforming urban governance models and achieving a low-carbon transition.

**Keywords:** smart city pilots; energy consumption; energy-saving effect; staggered DID; China

## 1. Introduction

Energy plays an important role in economic development. However, globally and especially in China, economic growth has still not been able to eliminate fossil energy consumption, which has caused environmental pollution and climate change problems. In response to the growing problem of environmental pollution and climate change, many countries have implemented a large number of energy conservation measures aimed at reducing fossil energy consumption and improving energy efficiency. Emerging digital technologies such as cloud computing, the Internet of Things, and big data have promoted low-carbon technology and fostered the development of new industries, which creates enormous space for energy conservation [1]. Many countries regard digital transformation as an essential direction to mitigate energy consumption. However, digital transformation may drive up energy consumption, because the application of digital technology is highly dependent on energy consumption [2].

As the largest energy consumer in the world, China's energy consumption reached 5.24 billion tons of standard coal in 2021, of which coal consumption accounted for 56%. The huge fossil energy consumption has contributed to air pollution and climate change problems, which are a major impediment to promoting sustainable development and achieving "30–60" carbon peak and neutrality goals. Meanwhile, China proposed the smart city pilots (SCP) policy in 2013 that aims to support the digital transformation of the economy through the integration of next-generation information technology. After implementing the SCP policy, the level of smart governance in Chinese cities has been significantly improved. It is essential to scientifically investigate the role of the SCP policy with regard to energy conservation, as this is important for promoting the green and low-carbon development of cities and achieving carbon peak and neutrality goals in time. In this context, this paper attempts to answer the following questions: Does the SCP policy promote the dual control of energy consumption? What is the underlying mechanism? Are the effects heterogeneous?

This paper makes the following contributions. First, the SCP policy is an important initiative to promote digital transformation, and there are in-depth studies on the SCP policy. However, more credible empirical support is needed to evaluate the impact of the SCP policy on energy consumption. This paper investigates the energy-saving effect of the SCP policy and expands relevant research in this field. Second, this paper uses city annual energy data from 2006 to 2019 and takes the SCP as a policy entry to estimate its energy-saving effect, which alleviates the problem of endogeneity and data limitation. Third, existing studies do not explain the mechanisms by which the SCP policy affects energy consumption and the differences in the impact of the SCP policy on energy consumption across geographic locations, resource endowments, and policy implementation times. This paper also explores the potential mechanisms by which the SCP policy affects energy consumption and examines how the heterogeneous effects vary with geographical location, resource endowments, and the timing of policy implementation.

## 2. Literature Review

### 2.1. Influencing Factors of Energy Consumption and Energy Intensity

An important prerequisite for achieving the goal of "carbon peaking and carbon neutrality" is the optimization of energy consumption and energy intensity. Numerous factors influence energy consumption and energy intensity. One of these is financial development. Lahiani analyzed the impact of financial development on renewable energy consumption in the United States between 1975Q1 and 2019Q4 and found that negative changes in overall and equity-based financial development measures had a significant impact on renewable energy consumption in the short term [3]. Chen et al. found that financial development has a significant negative impact on energy intensity in non-OECD countries, but a limited impact on energy intensity in OECD countries [4]. A second factor is infrastructure. Jain and Tiwari found that the optimization of transportation infrastructure is beneficial to reducing energy consumption in a sample of three cities: Delhi, Pune, and Patna [5]. Li et al. found that industrial transfer in China increases total energy consumption but reduces energy intensity [6]. Lin and Chen found that economic infrastructure construction will increase energy consumption and reduce energy intensity in the long term by using provincial data from China [7]. A third factor is innovation. Liu et al. concluded that an increase in innovation efficiency increases the regional energy consumption intensity in China [8]. The literature on the impact of urbanization on energy consumption is most relevant to the study in this paper, but the findings on this topic are inconsistent. Some of the research suggests that urbanization increases energy consumption. Using a sample of 22 emerging economies, Rafiq et al. found that urbanization significantly increased energy intensity [9]. Elliott argued that the direct effect of urbanization on energy intensity in China is positive [10]. Other scholars have provided empirical evidence that urbanization reduces energy consumption. In a sample of South Korea, Lee and Lim found that compact cities could help reduce energy consumption with optimal city size and effective policies with sufficient financial constraints [11]. Based on a sample of 193 Chinese cities, Lin and Zhu found that new-type urbanization had a significant energy-saving effect, with the effect being greater in resource-rich areas [12].

### 2.2. Effect of the SCP Policy

Smart cities are new urban development models that integrate new technologies such as artificial intelligence and edge computing into city construction and management, including multiple modules such as intelligent transportation systems, smart healthcare, smart grids, and smart governance [13]. More and more countries are using smart city construction as a means to alleviate urban economic development problems, especially in developing countries [14]. Most studies have focused on discussing the impact of smart city construction policies on innovation development, livelihood employment, life and health, etc. Based on Chinese data, Guo and Zhong offered supporting evidence for the positive effect of the SCP policy on urban innovation performance [15]. Luo et al. found

that the SCP policy significantly boosted employment and wages, although this effect was mainly concentrated in urban areas and the information industry [16]. However, some studies have also pointed out that smart cities have failed to achieve the expected effects. Using a sample from India, Mullick and Patnaik noted that smart city construction failed to improve the ability of cities to cope with epidemics due to the digital divide among citizens [17]. Some studies point out that smart city construction also plays a role in coping with environmental issues. Arsenio et al. used the city of Agueda, the first smart city in Portugal, as a sample, and the authors argued that smart city construction facilitates transportation decarbonization [18]. Wu argued that the SCP policy significantly reduces indirect household carbon emissions [19]. Ma and Wu also supported the conclusion that the SCP policy is beneficial to carbon emission reduction [20]. Furthermore, some scholars have found that smart city construction plays an important role in improving green total factor productivity [21–23]. As an important branch of environmental issues, very few scholars have focused on the impact of smart city construction on energy. Based on data from 251 cities in China from 2003 to 2016, Yu and Zhang found that the SCP policy had a significant positive effect on energy efficiency [21]. Guo and Wang found that smart city construction improves energy efficiency, and the innovation effect is a key mechanism [24].

*2.3. Literature Gap*

Although existing related research has made efforts to focus on the impact of smart cities and the factors influencing energy consumption, there are still some gaps in the literature. First, existing studies have not paid enough attention to the relationship between the SCP policy and energy consumption, although there is some literature that confirms the positive effect of the SCP policy on energy efficiency. However, more credible empirical support is needed to evaluate the impact of the SCP policy on energy consumption. Second, existing studies do not explain the mechanisms by which the SCP policy affects energy consumption and the differences in the impact of smart city construction on energy consumption across geographic locations, resource endowments, and policy implementation times. A few studies have analyzed the mechanism by which the SCP works mainly from the perspective of innovation effects, but discussions based on industrial structure effects are lacking.

## 3. Theoretical Analysis

The specific goals of the SCP policy in China include four aspects: smart construction and liveability, smart management, and smart industry and economy. Therefore, we argue that the SCP policy not only reduces energy consumption directly through wisdom management and wisdom construction, but it also indirectly influences energy consumption by promoting smart industry transformation and enhancing technological innovation.

The SCP policy may reduce energy consumption by improving smart construction and management. First, the SCP policy improves the level of intelligent management by building an efficient energy monitoring platform and decision support system to aid in decision making, thereby saving energy and improving energy efficiency. Building online energy consumption monitoring systems for high-energy-consuming enterprises using the Internet of Things and artificial intelligence technologies, as well as real-time monitoring systems for smart grids, can assist governments and enterprises in analyzing and forecasting energy consumption to improve energy-saving and efficiency strategies, which can effectively promote energy management and energy utilization efficiency. Second, the SCP policy is critical in promoting urban energy-saving projects. In the fields of buildings and infrastructure, IoT technology not only improves the intelligent management of buildings, but it also promotes the intelligent transformation of gas, heating systems, and lighting systems, thus effectively improving energy-saving effects. Third, smart transportation construction promotes the application of big data in transportation, and the transportation sector has effectively reduced urban congestion and energy consumption.

In addition, the SCP policy can affect energy consumption through smart industry transformation and low-carbon innovation.

The role of smart city construction in promoting industrial structure upgrading is widely acknowledged [25]. Smart city construction creates a huge demand and opportunity for the smart industry, providing a broader space for smart industry development. Smart technologies such as the IoT, big data, 5G, cloud computing, and AI are gradually replacing outdated traditional industries and promoting smart industry transformation. With the gradual development of emerging industries, humans, capital, energy, and other elements are achieving a readjustment from traditional industries to smart industries, which in turn promotes the low-carbon transformation of the urban industrial system. Meanwhile, with the deep integration of digitalization, traditional industries will be conducive to reducing energy consumption while saving production costs through transformation and upgrading, improving production efficiency, and achieving clean production.

Furthermore, the SCP policy has the potential to reduce energy consumption through low-carbon innovation. The SCP policy results in the clustering of innovation factors such as talent, enterprises, and research and development capital for smart industry development, which is conducive to the establishment of regional innovation systems, thereby comprehensively enhancing technological innovation. During the smart city construction process, the local government also attracts investment and promotes the development of smart industries in the region by matching financial funds. In addition, information enterprises also promote enterprise innovation by introducing advanced technologies and increasing R&D investment.

## 4. Methods and Data

### 4.1. Methodology

Currently, causal effect identification strategies based on potential causal models, such as instrumental variable (IV), DID, regression discontinuity (RD), and synthetic control (SCM), are becoming common research paradigms for empirical studies in the social sciences [26]. In contrast to other methods, the DID method has the advantages of being intuitive and easy to understand and operate, and it is more suitable for situations in which there are more pilot units. By 2019, 98 prefecture-level cities in the sample had been approved for the SCP, providing a good opportunity to estimate the energy-saving effect of the SCP policy using the staggered DID method. Specifically, we consider the 98 prefecture-level cities approved to carry out pilots to be the treatment group and the remaining cities that were not approved to be the control group. Furthermore, from 2013 to 2015, the Chinese Ministry of Housing and Urban–Rural Development (MOHURD) adopted a batch publication approach in selecting pilots, publishing three batches of pilot lists. Therefore, we built a staggered DID model, referring to the work in [27]. The model specifications are as follows:

$$Y_{it} = \alpha_0 + \alpha_1 SCP_{it} + \alpha_2 X_{it} + \mu_i + \nu_t + \varepsilon_{it} \tag{1}$$

In Equation (1), $Y_{it}$ represents the energy consumption level in city $i$ and in year $t$. $X_{it}$ represents variables. $\mu_i$ and $\nu_t$ represent the city and year fixed effects, respectively. $\varepsilon_{it}$ is the error term. $SCP_{it}$ represents the SCP policy, and $\alpha_1$ reflects the energy-saving effect of the SCP policy.

The common trend assumption, which states that the treatment and control groups must have a common trend, is required for using the staggered DID method. Following Beck et al. [28], we built a dynamic DID model:

$$Y_{it} = \alpha_0 + \sum_{k=-9, k \neq -1}^{k=6} \beta_k SCP_{it}^k + \delta X_{it} + \nu_t + \mu_i + \varepsilon_{it} \tag{2}$$

In Equation (2), $SCP_{it}^k$ equals one for city $i$ in the $k^{th}$ the year before or after implementing the SCP policy; otherwise, $SCP_{it}^k$ equals zero. We used one year before pilots as the

base period, which means that we exclude $SCP_{it}^{-1}$. $\beta_k$ are used to test the parallel trend assumption and to evaluate the dynamic effects of the SCP policy. When $k < 0$, the results of the estimated coefficients can be used to determine whether the sample satisfies the parallel trend hypothesis; meanwhile, when $k \geq 0$, the estimated coefficients can reflect the dynamic effects of the policy.

*4.2. Data and Variables*

We used energy consumption (LnE) and energy consumption intensity (LnEI) to measure the scale of the energy consumption level. Due to the lack of statistics, we estimated energy consumption through nighttime lighting data. Referring to the existing literature [29,30], we first calculate the total value of nighttime lights at the provincial and municipal levels based on the DMSP/OLS and NPP-VIIRS nighttime lights data. Then, we estimate the relationship between the total value of provincial nighttime lights and provincial energy consumption. The equation is as follows:

$$E_{it} = \alpha NLT_{it} + \varepsilon_{it} \tag{3}$$

In Equation (3), $E_{it}$ represents the energy consumption in $i$ province and in year $t$. $NLT_{it}$ represents the total value of nighttime lights in $i$ province and in year $t$. We estimate the $\alpha$ using a linear model without an intercept term.

Finally, we applied a top-down method to measure energy consumption at the city level in China by constructing a model of Equation (4):

$$E_i = E_k \times \left( \hat{E}_i / \hat{E}_k \right) \tag{4}$$

In Equation (4), $E_i$ is the estimated energy consumption at the city level. $E_k$ is the provincial energy consumption issued by the National Bureau of Statistics (NBS) of China. $\hat{E}_i$ and $\hat{E}_k$ are the estimated energy consumption of the city and province based on the total nighttime lighting values, respectively. In addition, we calculated the energy consumption intensity by dividing the total energy consumption by the real GDP.

We assigned values to the SCP according to the list of smart city pilots released by the Chinese MOHURD. When a city is selected as a smart city pilot, we assign the year and subsequent years to one, and otherwise, they are assigned to zero. In addition, to alleviate the effects of omitted variables, referring to Elliott et al. [10] and Xue and Zhou [31], we also used the share of secondary industry in GDP, foreign investment level, logarithm of total population, financial development level, and technology expenditure level as the controlled variables in the baseline model.

We collected data on the SCP from the official website of the MOHURD, which includes detailed information on the list of pilot cities and dates. Provincial energy consumption data came from the *China Energy Statistical Yearbook*. DMSP/OLS and NPP-VIIRS night lighting data came from NOAA's National Geophysical Data Center. Information on the socio-economic characteristics of the cities was obtained from the *China City Statistical Yearbook* for 2007 to 2020. Green patent applications were obtained from the Chinese Research Data Services Platform.

Table 1 presents the descriptive statistics. The data come from 224 prefecture-level cities and spans 12 years from 2006 to 2019. We used 2006 as the starting year to eliminate the effect of the Chinese government's policy of making energy consumption intensity a binding indicator in the five-year national economic and social development plan for the first time in 2006. The number of different observations changes because of the existence of statistical data.

**Table 1.** Descriptive statistics of key variables.

| Variable | Symbols | Obs. | Mean | Std.Dev. |
|---|---|---|---|---|
| Logarithm of energy consumption | Lne | 3324 | 6.858 | 0.839 |
| Logarithm of energy intensity | Lnei | 3324 | 9.252 | 0.785 |
| SCP | SCP | 3324 | 0.119 | 0.324 |
| Foreign investment level/% | Fdi | 3249 | 1.893 | 1.897 |
| Technology expenditure level/% | Rd | 3323 | 6.932 | 22.579 |
| Financial development level/% | Finance | 3321 | 0.814 | 0.507 |
| Logarithm of total population | Lnpop | 3324 | 5.855 | 0.669 |
| The share of secondary industry in GDP /% | Ind | 3319 | 49.088 | 10.686 |
| Logarithm of electricity consumption | Lnele | 2970 | 3.645 | 1.128 |
| Logarithm of electricity consumption intensity | Lnelei | 2970 | −3.127 | 0.742 |

Energy consumption intensity is measured by the ratio of total energy consumption to GDP. Foreign investment level is measured by the ratio of foreign direct investment to GDP. Technology expenditure level is measured by the proportion of government expenditure on science and technology to expenditure. Financial development level is measured by the ratio of financial loans to GDP.

## 5. Empirical Analysis

### 5.1. Main Results

We estimate Equation (1) to evaluate the energy effect of the SCP policy using a two-way fixed-effect model. Table 2 displays the results. Columns (1) and (2) in Table 2 do not include control variables, but columns (3) and (4) do. As shown in Table 2, the estimated coefficients of *SCP* are significantly negative at the 5% level, which indicates that the SCP policy exerts a positive influence on energy consumption and energy intensity. More specifically, the SCP policy reduces energy consumption and energy intensity by approximately 2.4% and 4.8% without control variables, and by nearly 3.3% and 5.3% with all control variables, respectively. This finding indicates that the cities save more energy when the SCP policy is implemented.

**Table 2.** Baseline results.

| Variables | Lne | Lnei | Lne | Lnei |
|---|---|---|---|---|
| | (1) | (2) | (3) | (4) |
| SCP | −0.024 ** | −0.048 *** | −0.033 *** | −0.053 *** |
| | (0.009) | (0.009) | (0.009) | (0.009) |
| Ind | | | 0.007 *** | 0.000 |
| | | | (0.001) | (0.001) |
| Fdi | | | −0.003 | −0.006 *** |
| | | | (0.002) | (0.002) |
| Lnpop | | | 0.238 *** | 0.106 ** |
| | | | (0.043) | (0.045) |
| Finance | | | −0.013 | 0.015 |
| | | | (0.010) | (0.010) |
| Rd | | | −1.282 *** | −2.598 *** |
| | | | (0.315) | (0.324) |
| Constant | 6.776 *** | 0.066 *** | 5.156 *** | −0.555 ** |
| | (0.003) | (0.003) | (0.255) | (0.263) |
| City fixed effect | Yes | Yes | Yes | Yes |
| Year fixed effect | Yes | Yes | Yes | Yes |
| Observations | 3136 | 3135 | 2922 | 2922 |
| R-squared | 0.968 | 0.946 | 0.973 | 0.949 |

The parentheses are standard errors. **, and *** indicate statistical significance levels at the 5%, and 1% levels, respectively. Lne indicates logarithm of energy consumption. Lnei indicates logarithm of energy intensity.

### 5.2. Parallel Trend Tests and Dynamic Effects

We next examine the common trend assumption for the staggered DID model by estimating Equation (2). Figure 1 plots the estimated coefficients in Equation (2), and the

dash lines represent 95% confidence intervals, highlighting two key points: the decline in energy consumption and energy intensity did not precede policy implementation, but it did occur soon after. As shown in Figures 1 and 2, the coefficients are insignificantly different from zero for all years before the implementation of the pilots, implying that the treatment and control groups show a common trend.

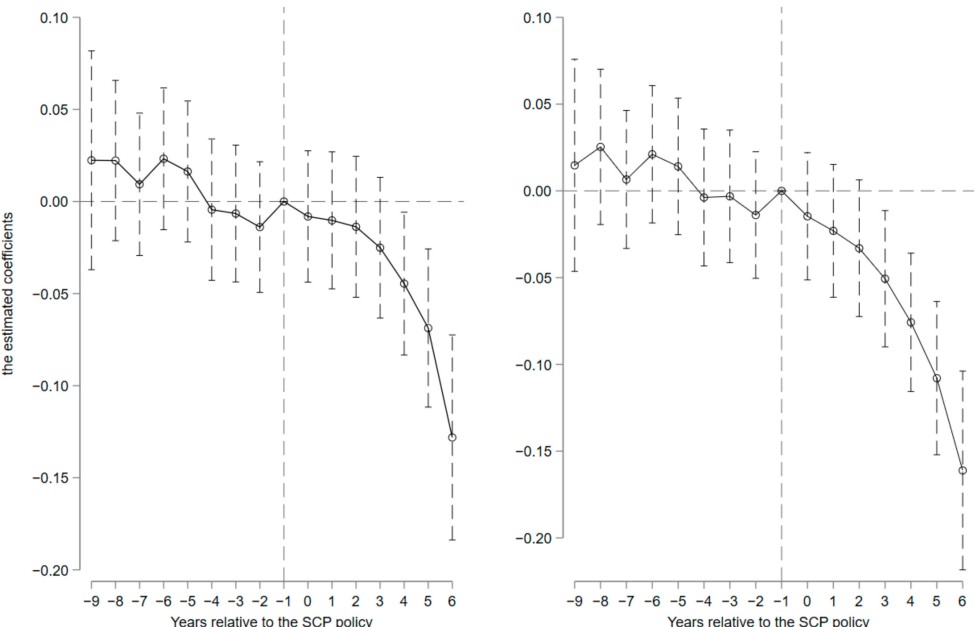

**Figure 1.** Parallel trend tests and dynamic effects. The left plot shows the results with the logarithm of total energy consumption as the explanatory variable, and the right plot shows the results with the logarithm of energy intensity as the explanatory variable. The vertical axis represents the estimated coefficient, and the horizontal axis indicates the relative time node of the SCP policy implementation.

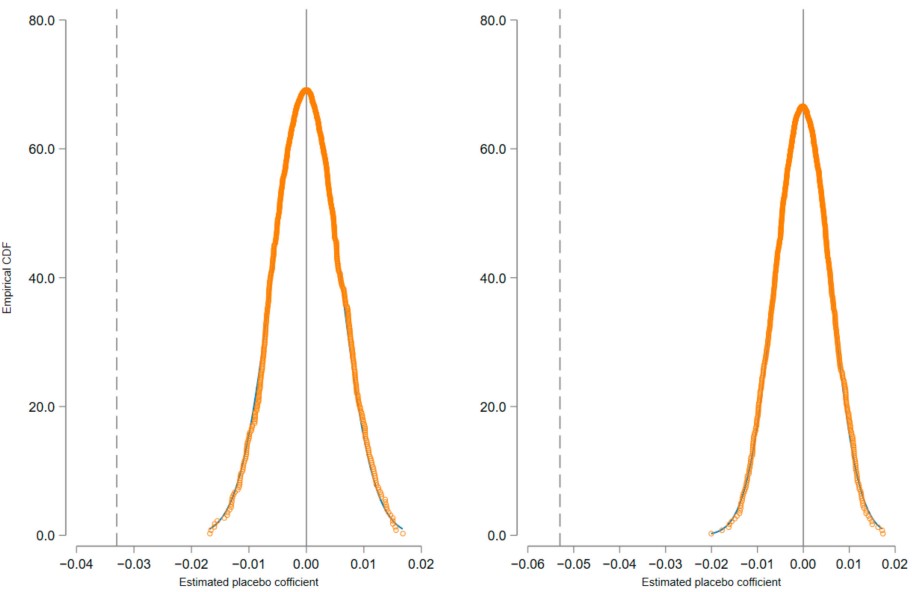

**Figure 2.** Placebo tests of logarithm of energy consumption. The left plot shows the results with the logarithm of total energy consumption as the explanatory variable, and the right plot shows the results with the logarithm of energy intensity as the explanatory variable. The orange circle is the probability of occurrence of estimated coefficients. The blue line is the fitting the cumulative distribution function line. The dashed line is the estimated result of the benchmark result.

Figure 1 also shows the dynamic effects of the SCP policy. Following this, energy consumption and energy intensity fall after the implementation of the pilots, such that the coefficients of $D_{it}^3 - D_{it}^6$ are negative and significant at the 10% level. In particular, the policy effect becomes significant in the third year of policy implementation and grows stronger as the year progresses. This finding indicates that the energy-saving effect of the SCP policy has a policy lag, but the policy effect starts to appear from the third year of the policy and increases as the year progresses.

### 5.3. Placebo Tests

Placebo tests, which randomize the control group, were employed to examine whether the results are influenced by potentially omitted variables. Theoretically, the coefficient expression $\hat{\beta}$ can be derived from Equation (1) as follows:

$$\hat{\beta} = \beta + \gamma \times \frac{\text{cov}(\text{SCP}_{it}, \varepsilon_{it} \,|\, W\,)}{\text{cov}(\text{SCP}_{it} \,|\, W\,)} \tag{5}$$

In Equation (5), $W$ includes all other control variables and fixed effects. $\gamma$ is the effect of unobserved factors on the explanatory variables. If $\gamma = 0$, then the unobserved factors do not affect the estimation results, which means $\hat{\beta}$ is unbiased, although this cannot be directly verified. Therefore, referring to previous studies [32–34], we employed a placebo test by randomizing the control group. Specifically, we randomly generate a list of pilots, thus generating a virtual estimate $\hat{\beta}^{random}$. This procedure is repeated 1000 times and correspondingly generates 1000 $\hat{\beta}^{random}$. Given the random data generation process, the virtual *SCP* variables should produce no significant estimates with a magnitude close to zero.

Figure 2 shows the results of the placebo tests. As shown, the distribution of $\hat{\beta}^{random}$ is centered around zero and follows a normal distribution, indicating that $\hat{\beta}^{random}$ does not affect energy consumption and energy intensity. Furthermore, we note that the benchmark estimators are outside the confidence interval of the distribution. In sum, the placebo tests indicate that the significantly positive effect of the SCP policy on energy consumption and energy intensity is not due to unobserved factors.

### 5.4. Other Robustness Tests

To ensure the accuracy of the results, we ran other robustness tests with the staggered DID model from Equation (1) as the benchmark.

#### 5.4.1. Excluding the Interference of Other Policies

Other energy policies implemented during the sample period could also confound the baseline estimation results. Therefore, we employed two strategies to exclude the interference of other policies.

First, we identified large city-level environmental policies that have been implemented since 2010, specifically the national comprehensive demonstration of energy-saving and emission reduction fiscal policy implemented since 2011, as well as the regional industrial green transformation development pilots that have been implemented since 2014. We included dummy variables for these policies in the equation, thus controlling for the effect of the relevant energy policies on the results. Columns (1) and (2) in Table 3 show that the coefficients of the SCP policy are more similar to the results of the baseline regression, indicating that the other energy policies do not cause bias in the estimation results.

**Table 3.** Robustness tests.

| Variables | (1) | (2) | (3) | (4) | (5) | (6) |
|---|---|---|---|---|---|---|
| SCP | −0.031 *** | −0.051 *** | −0.032 *** | −0.038 *** | −0.070 *** | −0.086 *** |
| | (0.009) | (0.009) | (0.009) | (0.009) | (0.025) | (0.025) |
| Ind | −0.004 * | −0.006 *** | 0.006 *** | 0.000 | 0.022*** | 0.015 *** |
| | (0.002) | (0.002) | (0.001) | (0.001) | (0.002) | (0.002) |
| Fdi | −1.272 *** | −2.597 *** | 0.011 *** | 0.012 *** | 0.004 | 0.001 |
| | (0.315) | (0.324) | (0.002) | (0.003) | (0.005) | (0.005) |
| Lnpop | −0.010 | 0.018 * | −0.060 | −0.139 *** | 0.244 ** | 0.126 |
| | (0.010) | (0.010) | (0.048) | (0.049) | (0.115) | (0.115) |
| Finance | 0.257 *** | 0.128 *** | 0.002 | 0.003 | 0.118 *** | 0.159 *** |
| | (0.044) | (0.045) | (0.012) | (0.012) | (0.034) | (0.034) |
| Rd | 0.006 *** | −0.000 | −1.597 *** | −2.445 *** | −1.178 | −2.493 *** |
| | (0.001) | (0.001) | (0.353) | (0.358) | (0.925) | (0.925) |
| Constant | 5.048 *** | −0.681 ** | 6.907 *** | 0.850 *** | 2.108 *** | −3.973 *** |
| | (0.257) | (0.264) | (0.286) | (0.290) | (0.693) | (0.693) |
| Other policies | Yes | Yes | No | No | No | No |
| City fixed effect | Yes | Yes | Yes | Yes | Yes | Yes |
| Year fixed effect | Yes | Yes | Yes | Yes | Yes | Yes |
| Province × year fixed effect | No | No | Yes | Yes | No | No |
| Observations | 2922 | 2922 | 2893 | 2893 | 1619 | 1619 |
| R-squared | 0.973 | 0.950 | 0.979 | 0.963 | 0.925 | 0.708 |

The parentheses are standard errors. *, **, and *** indicate statistical significance levels at the 10%, 5%, and 1% levels, respectively.

Second, even though the city and year fixed effects are included above, the sample is still subject to the problem of the influence of different time trends. Therefore, we further included province year fixed effects to capture the policy effects of each province and city over time, as shown in columns (3) and (4) in Table 3. The estimated coefficients of the SCP policy remain significantly negative, which undoubtedly supports the findings of this paper.

5.4.2. Substitute Explanatory Variables

We also performed a robustness test by replacing the indicators that measure the level of energy consumption. Specifically, we replace the energy consumption and energy intensity with society-wide electricity consumption and the intensity of this electricity consumption, respectively. Columns (5) and (6) in Table 3 show that the coefficients of the SCP policy remain negative after replacing the explanatory variables.

*5.5. Heterogeneity Analysis*

The preceding analysis demonstrates that the SCP policy can significantly reduce energy consumption and energy intensity. Is there a difference in the energy effect of the SCP policy for cities with different characteristics? To answer this, we further tested the heterogeneous effect in three ways. Tables 4–6 report the results of the heterogeneity analysis for different city types.

**Table 4.** Heterogeneity results for different regions.

| Variables | East | | Middle | | West | |
|---|---|---|---|---|---|---|
| | **Lne** | **Lnei** | **Lne** | **Lnei** | **Lne** | **Lnei** |
| | **(1)** | **(2)** | **(3)** | **(4)** | **(5)** | **(6)** |
| SCP | −0.098 *** | −0.093 *** | 0.007 | 0.021 | −0.049 *** | −0.073 *** |
| | (0.013) | (0.013) | (0.014) | (0.014) | (0.019) | (0.021) |
| Control variables | Yes | Yes | Yes | Yes | Yes | Yes |
| City fixed effect | Yes | Yes | Yes | Yes | Yes | Yes |
| Year fixed effect | Yes | Yes | Yes | Yes | Yes | Yes |
| Observations | 824 | 824 | 849 | 849 | 898 | 898 |
| R-squared | 0.986 | 0.978 | 0.976 | 0.961 | 0.960 | 0.909 |

The parentheses are standard errors. *** indicates statistical significance level at the 1% level.

**Table 5.** Heterogeneity results for different resource endowments.

| Variables | RB | | Non-RB | |
|---|---|---|---|---|
| | **Lne** | **Lnei** | **Lne** | **Lnei** |
| | **(1)** | **(2)** | **(3)** | **(4)** |
| SCP | −0.038 ** | −0.060 *** | −0.026 ** | −0.044 *** |
| | (0.015) | (0.016) | (0.011) | (0.011) |
| Control variables | Yes | Yes | Yes | Yes |
| City fixed effect | Yes | Yes | Yes | Yes |
| Year fixed effect | Yes | Yes | Yes | Yes |
| Observations | 1231 | 1231 | 1691 | 1691 |
| R-squared | 0.970 | 0.936 | 0.975 | 0.956 |

The parentheses are standard errors. **, and *** indicate statistical significance levels at the 5%, and 1% levels, respectively.

**Table 6.** Heterogeneity results for different batches.

| Variables | First Batch | | Second Batch | | Third Batch | |
|---|---|---|---|---|---|---|
| | **Lne** | **Lnei** | **Lne** | **Lnei** | **Lne** | **Lnei** |
| | **(1)** | **(2)** | **(3)** | **(4)** | **(5)** | **(6)** |
| SCP | −0.050 *** | −0.076 *** | −0.048 *** | −0.070 *** | −0.025 ** | −0.049 *** |
| | (0.013) | (0.013) | (0.010) | (0.010) | (0.011) | (0.011) |
| Control variables | Yes | Yes | Yes | Yes | Yes | Yes |
| City fixed effect | Yes | Yes | Yes | Yes | Yes | Yes |
| Year fixed effect | Yes | Yes | Yes | Yes | Yes | Yes |
| Observations | 2134 | 2134 | 2561 | 2561 | 2495 | 2495 |
| R-squared | 0.969 | 0.946 | 0.973 | 0.950 | 0.970 | 0.946 |

The parentheses are standard errors. **, and *** indicate statistical significance levels at the 5%, and 1% levels, respectively.

First, the sample was divided into eastern, central, and western cities, following Zheng and Li [35], and the results are reported in Table 4. Columns (1) and (2) in Table 4 report the effect for eastern cities, and the coefficients of SCP are −0.098 and −0.093. Columns (3) and (4) in Table 4 use a sample in central cities, and the coefficients are 0.007 and 0.021. The sample in columns (5) and (6) in Table 4 are western cities, with coefficients of −0.049 and −0.073, respectively, which indicates that the effect in western cities is stronger than in eastern cities, and the effect in central cities is the weakest.

Second, we categorized the sample as resource-based (RB) and non-RB based on the "National Plan for Sustainable Development Plan of Resource-based Cities of China (2013–2020)". Table 5 reports the heterogeneity results for different resource endowments.

The samples in columns (1) and (2) are RB cities, with coefficients of −0.038 and −0.060, respectively. The samples in columns (3) and (4) are non-RB cities, with coefficients of −0.026 and −0.044, respectively. Furthermore, all coefficients are significant at the 1% level. The results in Table 5 show that the SCP policy has a positive energy-saving effect in both the RB city group and the non-RB city group, but it has a greater role in RB cities.

　　　Third, the sample was divided into three groups based on the batches established by the pilot cities. The first batch of pilots, which included 32 cities, was established in January 2013. The second batch was piloted in August 2013 and included 41 cities. The third batch was piloted in April 2015 and included 28 cities. Table 6 presents the heterogeneity results for the different batches. Columns (1) and (2) show that in the first batch of pilot cities, energy consumption and energy intensity were significantly reduced. Similarly, columns (3) and (4) show that the coefficients of the second batch are smaller than those of the first batch. Unlike the results for the first and second batches, the third batch in columns (5) and (6) shows that the effects on energy consumption and energy intensity are not significant. The effects of the SCP policy on energy consumption and energy intensity often have a two-year policy lag, which may have resulted in the insignificant policy effects for the third pilot batch.

### 5.6. Mechanisms

　　　We next sought to elucidate the specific influence mechanism of this effect. According to the policy implementation details of the Smart City Pilot Program, the SCP policy not only directly reduces energy consumption and energy intensity, but it also exerts an influence through low-carbon innovation and smart industry transformation. Therefore, following Baron and Kenny [36], we comprehensively examined the specific influence mechanisms of the SCP policy in the reduction of energy consumption and energy intensity. Specifically, we first examined the effect of the SCP policy on low-carbon innovation and smart industry transformation, as follows:

$$M_{it}^{s} = \alpha_0 + \alpha_1 SCP_{it} + \alpha_2 X_{it} + \mu_i + v_t + \varepsilon_{it} \tag{6}$$

　　　Based on Equation (6), we include the mechanism variables in Equation (1) to verify the two main effects of the SCP policy on energy consumption and energy intensity:

$$Y_{it} = \alpha_0 + \alpha_1 SCP_{it} + \alpha_2 X_{it} + \beta_s M_{it}^{s} + \mu_i + v_t + \varepsilon_{it} \tag{7}$$

　　　In Equations (6) and (7), $M_{it}^{s}$ represents the mechanism variable that contains two variables: low-carbon innovation (LI) and smart industry transformation (SIT). We use the logarithm of the number of patents filed for low-carbon inventions (LI1) and the logarithm of the number of low-carbon utility model patent applications (LI2) to measure low-carbon innovation LI, respectively. Furthermore, we use the ratio of total telecom business to GDP (SIT1) and the ratio of total telecom business to the value-added of tertiary industry (SIT2) measure SIT. The meanings of the other variables are the same as in Equation (1).

　　　Columns (1)–(4) in Table 7 present the results of the effect of the SCP policy on LI and SIT by estimating Equation (4). As shown in column (1) and (2), the coefficient of *SCP* is positive but not significant, which means that the SCP policy does not improve low-carbon innovation. It is evident that the SCP policy focuses on application, rather than technological innovation, and thus the impact on LI is not significant. The results in column (3) and (4) show that the coefficient of *SCP* is significantly positive, indicating that the SCP policy improves SIT in the pilot cities.

**Table 7.** Mechanism tests.

| Variables | LI1 | LI2 | SIT1 | SIT2 | Lne | Lnei | Lne | Lnei |
|---|---|---|---|---|---|---|---|---|
| | (1) | (2) | (3) | (4) | (5) | (6) | (7) | (8) |
| SCP | 0.040 | −0.008 | 0.002 * | 0.007 ** | −0.033 *** | −0.054 *** | −0.032 *** | −0.053 *** |
| | (0.034) | (0.031) | (0.001) | (0.003) | (0.009) | (0.009) | (0.009) | (0.009) |
| SIT1 | | | | | −0.194 * | −0.116 * | | |
| | | | | | (0.154) | (0158) | | |
| SIT2 | | | | | | | −0.120 ** | −0.097 * |
| | | | | | | | (0.060) | (0.062) |
| Control variables | Yes | Yes | Yes | Yes | Yes | Yes | Yes | Yes |
| City fixed effect | Yes | Yes | Yes | Yes | Yes | Yes | Yes | Yes |
| Year fixed effect | Yes | Yes | Yes | Yes | Yes | Yes | Yes | Yes |
| Observations | 2909 | 2909 | 2904 | 2904 | 2904 | 2904 | 2904 | 2904 |
| R-squared | 0.940 | 0.936 | 0.412 | 0.421 | 0.973 | 0.950 | 0.973 | 0.950 |

The parentheses are standard errors. *, **, and *** indicate statistical significance levels at the 10%, 5%, and 1% levels, respectively.

We next examined the effect of SIT on energy consumption and energy intensity by estimating Equation (5). Columns (5)–(8) in Table 7 show that the coefficient of SIT is significantly negative when total energy or energy intensity is the explanatory variable, whereas the coefficient of *SCP* remains significant, and the absolute value of the coefficient decreases slightly compared with the baseline results, indicating that smart industrial transformation has a significant inhibitory effect on energy and intensity.

The possible reason for this phenomenon is that smart city construction creates a huge demand and opportunity for the smart industry, providing a broader space for smart industry development. Smart industries such as IoT, big data, 5G, cloud computing, and AI are gradually replacing outdated traditional industries and promoting smart industry transformation [37]. With the gradual development of emerging industries, humans, capital, energy, and other elements are achieving a readjustment from traditional industries to smart industries, which in turn promotes the low-carbon transformation of the urban industrial system, improves production efficiency, and realizes clean production, which will help to save production costs while reducing energy consumption [38]. In sum, the above results indicate that the SCP policy mainly promotes the dual control of energy consumption and energy intensity by promoting smart industrial transformation.

## 6. Conclusions and Implications

This paper has employed the staggered DID method to examine the effect of the SCP policy on energy consumption and energy intensity, and the underlying mechanism and heterogeneous effect were analyzed based on China's smart city pilot initiative and panel data of 224 prefecture-level cities from 2006 to 2019. The primary conclusions are as follows. First, the SCP policy has promoted the dual control of energy consumption and energy intensity. Specifically, the SCP policy significantly reduced energy consumption and energy intensity by 3.3% and 5.3%, respectively. These results were still reliable after a common trend test, a placebo test, and other robustness tests. Second, heterogeneity analysis revealed that compared with the middle and eastern, non-RB, and later pilot cities, the SCP policy has exerted a stronger impact on the western, RB, and pioneering pilot cities. Third, the results of the mechanism analysis showed that structural transformation was the main transmission mechanism, and there was insufficient evidence to suggest that the mechanism of technological innovation effect held.

Based on the above empirical conclusions, the following policy suggestions are offered.

First, this paper finds that the SCP policy is conducive to reducing energy consumption and intensity. Therefore, in the context of "carbon peak and carbon neutralization" and "digital China", the government should focus on promoting the construction of new smart cities. It should continue to increase investment in new-generation information and com-

munication technology, promote the application of information technology in sustainable urban development, and promote the deep integration of new-generation information technology with the real economy to drive the low-carbon development of cities.

Second, this paper has found that the SCP policy has a heterogeneous impact on energy consumption and intensity. Therefore, the central and local governments need to fully consider the characteristics of urban energy resource endowment and implement top-level design for the SCP policy according to local conditions to avoid a "one-size-fits-all" approach.

Third, the results of the mechanism analysis showed that the main effect of the SCP policy came from the transformation of industrial structure, whereas the role of technological innovation was relatively limited. This suggests that the government should achieve low-carbon urban development through the development of strategic new industries based on digital intelligence technologies such as the IoT and cloud computing. At the same time, when designing and planning smart city policies, there should be more explicit planning and design to promote technological innovation, for example, by building enterprises and research institutions to establish innovation platforms.

This paper still has the following constraints: (1) This paper evaluates the energy-saving effect of the SCP policy at the macro-city level. It is necessary to further investigate this problem when micro firm-level data are available. (2) Constrained by the staggered DID model, we cannot discuss the spatial effects of the SCP policy. In future research, we can use the spatial DID model to further investigate the effect of the SCP policy on the energy consumption of surrounding cities.

**Author Contributions:** Conceptualization, F.X. and M.Z.; methodology, F.X.; software, F.X. and J.L.; validation, F.X. and M.Z.; formal analysis, F.X.; investigation, F.X.; resources, F.X.; data curation, F.X.; writing—original draft preparation, F.X.; writing—review and editing, F.X. and J.L.; visualization, F.X.; supervision, M.Z.; project administration, M.Z.; funding acquisition, M.Z. All authors have read and agreed to the published version of the manuscript.

**Funding:** This research was funded by Basic Research Project of Chinese Academy of Social Sciences [2022], Industrial and Regional Think Tank Project of Chinese Academy of Social Sciences [2020G02].

**Institutional Review Board Statement:** Not applicable.

**Informed Consent Statement:** Not applicable.

**Data Availability Statement:** Publicly available datasets were analyzed in this study. Provincial energy consumption data came from the *China Energy Statistical Yearbook*. DMSP/OLS and NPP-VIIRS night lighting data came from NOAA's National Geophysical Data Center. Other data came from the *China City Statistical Yearbook* and the Chinese Research Data Services Platform.

**Conflicts of Interest:** The authors declare no conflict of interest.

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
