# Peer review of "Are Cities Saving Energy by Getting Smarter? Evidence from Smart City Pilots in China"

_sustainability, doi:10.3390/su15042961_

Round 1

Reviewer 1 Report

This paper examine the effect of the SCP policy on energy consumption and energy intensity, and the underlying mechanism and heter-ogeneous effect are analyzed. 

1. The Introduction should be condensed, and it is enough to explain the relevant research background.

2.In the literature review section, the content of relevant research on SCP is simple, which makes it difficult for readers to understand the importance of this paper. The focus of this part should be strengthened, and the research innovation of this paper should be highlighted, not a data analysis report.

3.The Mechanisms of the this paper should be concise and to the point, that is, clearly describe the correlation and influence relationship between China's SCP policy and smart construction and liveability, smart management, and smart industry and economy.

4.The detailed information on the list of pilot cities and dates, provincial energy con-sumption data, DMSP/OLS and NPP-VIIRS night lighting data, information on the socio-economic characteristics of the cities and Green patent applications should be provided in the form of supplementary materials, if applicable.

5.Although the results have undergone robustness tests such as common trend tests and placebo tests, the shortcomings of the above-mentioned test methods should still be provided based on the data in this paper.

6.Figures 1 to 4 should provide a clearer picture. And because the content of the picture is simple, it is recommended to merge.

7.Typical city cases should be provided in the heterogeneity analysis to support the analysis of the results.

8.The conclusion section should present the findings of the study, and a detailed discussion of related extensions should be presented in the analysis section.

Author Response

Thank you so much for taking your busy time to review our revised manuscript. We believe that these remarks are very excellent and insightful for our research. Thanks very much again for your detailed and thorough review. Also, we believe that your further comments and suggestions are very helpful for our research. We humbly read each of your suggestions. Following your great comments, we have thoughtfully made the corresponding changes and give corresponding explanations of what we have changed from point by point outlined below. We hope we have addressed all of your concerns. We are deeply grateful for all your comments.

For the sake of presentation, the comments of the reviewers are numbered and duplicated in italics, and our responses are given in plain. All changes made to the manuscript are marked in yellow color. The page and line numbers of revised texts in our responses refer to our revised manuscript.

  1. The Introduction should be condensed, and it is enough to explain the relevant research background.

Answer 1: Thank you very much for your valuable comments and suggestions. We have condensed the introduction section as much as possible. Specifically, we have combined the first and second paragraphs of the introduction and removed unnecessary content.

2.In the literature review section, the content of relevant research on SCP is simple, which makes it difficult for readers to understand the importance of this paper. The focus of this part should be strengthened, and the research innovation of this paper should be highlighted, not a data analysis report.

Answer 2: Thank you very much for your advice. We have reorganized the literature review and added a large amount of new literature to highlight more of the innovative points of this paper. In addition, we have created three subsections to make the literature review more logical. In the first subsection, we focus on the factors influencing energy consumption and energy intensity, in the second subsection we summarize the literature on the effect of the SCP policy, and in the third subsection, we add to the existing literature gaps.

3.The Mechanisms of the this paper should be concise and to the point, that is, clearly describe the correlation and influence relationship between China's SCP policy and smart construction and liveability, smart management, and smart industry and economy.

Answer 3: Thank you very much for your advice. In the revised draft, we have removed the section “mechanism” and simplified the mechanism analysis as much as possible.

4.The detailed information on the list of pilot cities and dates, provincial energy consumption data, DMSP/OLS and NPP-VIIRS night lighting data, information on the socio-economic characteristics of the cities and Green patent applications should be provided in the form of supplementary materials, if applicable.

Answer 4: Most of the data is from publicly available data. We collected data on the SCP from the official website of the MOHURD, which includes detailed information on the list of pilot cities and dates. Provincial energy consumption data come from China Energy Statistical Yearbook. DMSP/OLS and NPP-VIIRS night lighting data come from NOAA’s National Geophysical Data Center. Information on the socio-economic characteristics of the cities was obtained from the China Urban Statistical Yearbook for 2007 to 2020. Green patent applications were obtained from the Chinese Research Data Services Platform. We have organized the data into dta format.

5.Although the results have undergone robustness tests such as common trend tests and placebo tests, the shortcomings of the above-mentioned test methods should still be provided based on the data in this paper.

Answer 5: Thank you for your question. We have done a series of robustness tests such as common trend tests and placebo tests to demonstrate that the SCP policy could reduce energy consumption and intensity. Therefore, it's difficult to argue that there are shortcomings of these robustness testing methods. Of course, the limitations of the different method prevent us from discussing some interesting issues. We have also added a discussion of the limitations of this paper in Section 5 of the revised manuscript.

6.Figures 1 to 4 should provide a clearer picture. And because the content of the picture is simple, it is recommended to merge.

Answer 6: Thank you very much for your advice. We have merged Figure 1 and Figure 2 into Figure 1, and Figure 3 and Figure 4 into Figure 2. In addition, we provide explanatory notes on the new figures.

7.Typical city cases should be provided in the heterogeneity analysis to support the analysis of the results.

Answer 7: Thank you very much for your valuable comments and suggestions. Although we would love to add some case discussions, however it is very unfortunate that the heterogeneity analysis section still examines the average treatment effects over different groups and we have difficulty getting the energy-saving effects of the SCP policy for a specific city.

8.The conclusion section should present the findings of the study, and a detailed discussion of related extensions should be presented in the analysis section.

Answer 8: Thank you very much for your advice. We add the analysis of the empirical results from the theory, and in addition, we add and improve the results of the concluding part of the study.

Reviewer 2 Report

The authors attempted to investigate the effect of smart city pilots (SCP) on energy consumption features in China. This is not a new topic as extensive studies have explored this research question from various perspectives. Even though the authors generally adopted a rigorous methodology to reveal such a policy effect, I have the following concerns that the authors must address:

(1) Why would the authors like to discuss such an old topic? You must focus on specific reasons empirically or theologically.

(2) Please modify your literature review accordingly. It only looks like a simple “stack-up” and the conclusion is very farfetched. Previous studies certainly concern about the issue of endogeneity and the topic of energy consumption.

(3) Please put sections 2 and 3 together. It is not necessary to have an independent section “mechanism”.

(4) The section of methodology is strange. The demonstration is quite inconsistent with the equations. What is pollution and NFC? How could you use the data of DMSP/OLS and NPP-VIIRS night lighting? It appears that the authors left something from a previous version of this article or something else and forgot to change them.

(5) Most importantly, the effect of a policy is difficult to be discussed based on a static model. For instance, it would take a time for local governments to understand and implement the SCP policy, thus the size of this effect must be dynamic. I suggest the authors to use dynamic models to capture this pattern, in addition to a number of simple fixed effects models where SCP is assumed to have a value. You mentioned a model of dynamic DID, but you have not displayed the results.

(6) Should the authors focus on the discussion regarding DID? So far, I do not see such a consistency between your article title & abstract and discussion. Is the section 5.2 your main analysis? You should discuss much more in this regard.

(7) The conclusions are proverbial. The authors should thoroughly discuss and conclude how your findings add values to the present research domain.

Author Response

Thank you very much for reviewing the revised manuscript. We believe that your further comments and suggestions are highly constructive and very helpful for our research. Thanks very much again for your detailed and thorough review. We have thoughtfully taken into these comments and responded to your constructive suggestions from point by point outlined below. We hope we have addressed all of your concerns.

  1. Why would the authors like to discuss such an old topic? You must focus on specific reasons empirically or theologically.

Answer 1: Thank you very much for your problem. To the best of our knowledge although existing related research literature has made efforts to focus on the effect effects of smart cities and the factors influencing energy consumption, there are still some gaps in the literature. First, existing studies have not paid enough attention to the relationship between the SCP policy and energy consumption, although there is literature confirming the positive effect of the SCP policy on energy efficiency. However, more credible empirical support is needed for the impact of the SCP policy on energy consumption and energy mix. Second, existing studies do not explain the mechanisms by which the SCP policy affects energy consumption and the differences in the impact of smart city construction on energy consumption across geographic locations, resource endowments, and policy implementation times.

2.Please modify your literature review accordingly. It only looks like a simple “stack-up” and the conclusion is very farfetched. Previous studies certainly concern about the issue of endogeneity and the topic of energy consumption.

Answer 2: Thank you very much for your advice. We have reorganized the literature review and added a large amount of new literature so as to highlight more the innovative points of this paper. In addition, we have created three subsections in order to make the literature review more logical. In the first subsection we focus on the factors influencing energy consumption and energy intensity, in the second subsection we summarize the literature on the effect of the SCP policy, and in the third subsection we add to the existing literature gaps.

3.Please put sections 2 and 3 together. It is not necessary to have an independent section “mechanism”.

Answer 3: Thank you very much for your advice. We have removed section “mechanism” according to your suggestion. In addition, we have merged parts of the Section 3 to Section 2, and the parts involving theoretical analysis into the Section 4 “empirical analysis”.

4.The section of methodology is strange. The demonstration is quite inconsistent with the equations. What is pollution and NFC? How could you use the data of DMSP/OLS and NPP-VIIRS night lighting? It appears that the authors left something from a previous version of this article or something else and forgot to change them.

Answer 4: Thank you very much for your valuable suggestions. We have modified equations (1) and (2), and restated the meaning of the relevant variables. In addition, we have added how to estimate energy consumption using data from DMSP / OLS and NPP-VIIRS nighttime lighting in lines 172-187 of the revision.

5.Most importantly, the effect of a policy is difficult to be discussed based on a static model. For instance, it would take a time for local governments to understand and implement the SCP policy, thus the size of this effect must be dynamic. I suggest the authors to use dynamic models to capture this pattern, in addition to a number of simple fixed effects models where SCP is assumed to have a value. You mentioned a model of dynamic DID, but you have not displayed the results.

Answer 5: Thank you very much for your advice. Following Beck et al. (2010), we evaluated the dynamic effects of SCP policy by using a dynamic DID model. Figure 1 reports the relevant regression results. In Figure 1, when , the results of the estimated coefficients can be used to determine whether the sample satisfies the parallel trend hypothesis; while when , the estimated coefficients can reflect the dynamic effects of the policy. As shown in Figure 1, it can be seen that the energy-saving effect of smart cities has a policy lag, but the policy effect starts to appear from the third year of policy implementation and grows stronger as the years go by.

6.Should the authors focus on the discussion regarding DID? So far, I do not see such a consistency between your article title & abstract and discussion. Is the section 5.2 your main analysis? You should discuss much more in this regard.

Answer 6: You are right. The empirical analysis part is the focus of this paper. And thank you very much for your advice. The section 4 Empirical Analysis is the focus of this paper. We expand this section with more discussion of the empirical results to make it more convincing.

7.The conclusions are proverbial. The authors should thoroughly discuss and conclude how your findings add values to the present research domain.

Answer 7: Thank you very much for your problem. Based on the empirical analysis, we have further added the interpretation of the energy-saving effect of the SCP policy. In addition, the possible contributions of this paper are the following three points: First, the SCP policy is an important initiative to promote digital transformation, and there are in-depth studies on the SCP policy. This paper investigates the energy-saving effect of the SCP policy and expands relevant research in this field. Second, this paper constructs city-annual energy data from 2006 to 2019, and takes the SCP policy as a policy entry to estimate the energy-saving effect of the SCP policy, which alleviates the problem of endogeneity and data limitation. Third, this paper also explores the potential mechanisms and examines how the heterogeneous effects vary with the geographical location, resource endowment, and timing of policy implementation.

Reviewer 3 Report

The article denotes a deficient development and must be significantly improved in all its sections. In the introduction, the contributions presented are different from what is presented. State-of-the-art needs to be more developed. It is limited in its scope and the references presented are limited in number and essentially of a regional character (from China). It would be interesting to present existing cases of application of the selected methodology, as well as the application of alternative methodologies and the justification for the final choice.

Equations should be checked and better explained.

The analysis of the results and the conclusions can also be significantly improved.

Author Response

Thank you so much for taking your busy time to review our revised manuscript. We believe that these remarks are very excellent and insightful for our research. Thanks very much again for your detailed and thorough review. Also, we believe that your further comments and suggestions are very helpful for our research. We humbly read each of your suggestions. Following your great comments, we have thoughtfully made the corresponding changes and give corresponding explanations of what we have changed from point by point outlined below. We hope we have addressed all of your concerns. We are deeply grateful for all your comments.

For the sake of presentation, the comments of the reviewers are numbered and duplicated in italics, and our responses are given in plain. All changes made to the manuscript are marked in yellow color. The page and line numbers of revised texts in our responses refer to our revised manuscript.

  1. The article denotes a deficient development and must be significantly improved in all its sections. In the introduction, the contributions presented are different from what is presented. State-of-the-art needs to be more developed. It is limited in its scope and the references presented are limited in number and essentially of a regional character (from China). It would be interesting to present existing cases of application of the selected methodology, as well as the application of alternative methodologies and the justification for the final choice.

 Answer 1: Thank you for your question. Based on your suggestions, we have made substantive changes to all chapters. In addition, we have rewritten the literature review section and divided it into three subsections for combing. Meanwhile, we added some literature from countries outside of China.

Currently, causal effect identification strategies based on potential causal models, such as instrumental variables (IV), DID, regression discontinuity (RD), and synthetic control (SCM), are becoming common research paradigms for empirical studies in the social sciences. In this paper, the DID method is used to assess the energy-saving effects of the SCP policy. This is because the DID method has the advantages of being intuitive, easy to understand, and easy to operate, and is more suitable for situations with a large number of pilot units, and is widely used. Therefore, We also added a comparison and selection of research methods.

  1. Equations should be checked and better explained.

 Answer 2: Thank you very much for your valuable suggestions. We have modified equations (1) and (2), and restated the meaning of the relevant variables.

  1. The analysis of the results and the conclusions can also be significantly improved.

Answer 3: Thank you very much for your advice. We add the analysis of the empirical results from the theory, and in addition, we add and improve the results of the concluding part of the study.

Round 2

Reviewer 1 Report

Congratulations, the questions related to the manuscript have been basically resolved, and it is recommended to accept and publish.

Author Response

We sincerely thank you for your suggestions and recognition. Under your guidance, the quality of our articles has been significantly improved. Thank you again for your guidance!

Reviewer 2 Report

I still have some concerns about this manuscript. Please pay attention. You have to show your efforts to the reviewers.

(1) You mentioned many contributions of this study in the letter to reviewers so they also should be included in the main text, for example, in the section of introduction.

(2) The structure of this study is still less clear. Why don’t you develop a clear research question like how SCP policy affects energy consumption in the context of China is less empirically explored, thus based on both static and dynamic perspectives, the direct and indirect policy effect on energy consumption is discussed. There is no need to have a section of mechanism separately.

(3) In lines 368-377, please provide references.

(4)Finally, the authors must thoroughly discuss how the implication of SCP affect energy consumption in multiple ways based on the present studies. What are consistent/inconsistent? Have you explored a new context or expanded the existing theoretical framework? The literatures the authors cited are insufficient; at least 10-15 more are needed.

Author Response

To the Reviewer

Thank you so much for taking your busy time to review our revised manuscript. We believe that these remarks are very excellent and insightful for our research. Thanks very much again for your detailed and thorough review. Also, we believe that your further comments and suggestions are very helpful for our research. We humbly read each of your suggestions. Following your great comments, we have thoughtfully made the corresponding changes and given corresponding explanations of what we have changed from the point-by-point outlined below. We hope we have addressed all of your concerns. We are deeply grateful for all your comments.

(1) You mentioned many contributions of this study in the letter to reviewers so they also should be included in the main text, for example, in the section of introduction.

Answer 1: Thank you very much for your valuable comments and suggestions. The possible contributions of this paper are expressed in the last paragraph of the introduction. In addition, we have added to the contributions based on your suggestions.

(2) The structure of this study is still less clear. Why don’t you develop a clear research question like how SCP policy affects energy consumption in the context of China is less empirically explored, thus based on both static and dynamic perspectives, the direct and indirect policy effect on energy consumption is discussed. There is no need to have a section of mechanism separately.

Answer 2: Thank you very much for your suggestions. In fact, the paper focuses on the following issues: Does the SCP policy promotes the dual control of energy consumption? What is the underlying mechanism? Are the effects heterogeneous? Based on your suggestion, we have clearly stated in the introduction several issues to be discussed in this paper. For more details, see lines 46-47.

My co-authors and I discussed that mechanism analysis remains an important part of this paper, which is important for understanding how the SCP policy affect energy consumption. We believed that policies such as the Smart City Pilot are essentially just an implementation program, and that there is still a question about how the Smart City Pilot policy will affect energy consumption. Therefore, this paper attempts to explore this issue through a mechanism analysis.

(3) In lines 368-377, please provide references.

Answer 3: Thank you for your question. We have added references to this section. Among them, lines 368-377 refer to Ref. [36], and lines 377-391 refer to Ref. [37].

(4) Finally, the authors must thoroughly discuss how the implication of SCP affect energy consumption in multiple ways based on the present studies. What are consistent/inconsistent? Have you explored a new context or expanded the existing theoretical framework? The literatures the authors cited are insufficient; at least 10-15 more are needed.

Answer 4: Thank you very much for your question and suggestions. We combed through the recently published articles, and we found that existing studies mainly examine the mechanisms of smart cities affecting energy consumption based on the innovation effect perspective, but lack discussion of the industrial structure effect. In this paper, we examine two perspectives, the innovation effect and the industrial structure effect, and our study finds that the industrial structure effect is the main channel through which the SCP policy affect energy consumption. We discuss this question in the literature review section. For more details, see lines 128-130. In addition, Based on your suggestions, we have added 10 articles. Refer to the modified version for details.